# Impact of higher-income countries on child health in lower-income countries from a climate change perspective. A case study of the UK and Malawi

Eilish Hannah [1], Rachel Etter-Phoya [1,2]*, Marisol Lopez[1], Stephen Hall[3,4], Bernadette O'Hare[1]

1 School of Medicine, University of St Andrews, St Andrews, United Kingdom, 2 Tax Justice Network, Lilongwe, Malawi, 3 School of Economics, University of Leicester, Leicester, United Kingdom, 4 University of Pretoria, Pretoria, South Africa

* rmep1@st-andrews.ac.uk

## Abstract

Climate change is the number one threat to child health according to the World Health Organisation. It increases existing inequalities, and lower-income countries are disproportionately affected. This is unjust. Higher-income countries have contributed and continue to contribute more to climate change than lower-income countries. This has been recognised by the United Nations Committee on the Rights of the Child, which has ruled that states can be held responsible if their carbon emissions harm child rights both within and outside their jurisdiction. Nevertheless, there are few analyses of the bilateral relationship between higher- and lower-income countries concerning climate change. This article uses the UK and Malawi as a case study to illustrate higher-income countries' impact on child health in lower-income countries. It aims to assist higher-income countries in developing more targeted policies. Children in Malawi can expect more food insecurity and reduced access to clean water, sanitation, and education. They will be more exposed to heat stress, droughts, floods, air pollution and life-threatening diseases, such as malaria. In 2019, 5,000 Malawian children died from air pollution (17% of under-five deaths). The UK needs to pay its 'fair share' of climate finance and ensure adaptation is prioritised for lower-income countries. It can advocate for more equitable and transparent allocation of climate finance to support the most vulnerable countries. Additionally, the UK can act domestically to curtail revenue losses in Malawi and other lower-income countries, which would free up resources for adaptation. In terms of mitigation, the UK must increase its nationally determined commitments by 58% to reach net zero and include overseas emissions. Land use, heating systems and renewable energy must be reviewed. It must mandate comprehensive scope three emission reporting for companies to include impacts along their value chain, and support businesses, multinational corporations, and banks to reach net zero.

**Data Availability Statement:** This study is literature based, an online modelling tool has been used for some data. Information about this can be

found here https://medicine.st-andrews.ac.uk/grade/research/.

**Funding:** Funding for this work comes from NHS Education Scotland (EH) and The Professor Sonia Buist Global Child Health Research Fund (BOH). NHS Education Scotland pays the salary for Dr Eilish Hannah. The funders had no role in study design, data collection and analysis, decision to publish, or preparation of the manuscript.

**Competing interests:** The authors have declared that no competing interests exist.

## Introduction

Eight of the ten hottest years on record have occurred in the last decade, and the World Health Organisation (WHO) has highlighted climate change as the number one threat to children's right to health [1,2]. Globally, nearly all children are at risk of the impacts of climate change, and children in low- and lower-middle-income (lower-income) countries will bear the brunt [3].

With current mitigation strategies, global temperatures in 2100 are predicted to be 2.8 degrees Celsius hotter than preindustrial levels [4]. We are not on track to limit the global temperature rise to below 1.5 degrees Celsius, and the probability of limiting it to below 2 degrees Celsius is 5% [4]. This will lead to a substantial increase in temperature volatility driving adverse health and economic impacts [4] The International Monetary Fund (IMF) estimated these costs were US$5.2 trillion in 2017 alone [5].

Children will be disproportionately impacted because their organs are still developing, meaning they are more susceptible to the impacts of climate change, such as high temperatures, air pollution, and diseases associated with a changing climate, such as malaria. They are also more vulnerable to extreme weather events, such as flooding, due to their size and developmental stage [6]. Thus, climate change affects their right to health in multiple interlinked ways. Helldén et al. summarised this as direct effects, such as temperature changes, precipitation and floods, droughts and wildfires, and indirect effects, such as changing vector patterns of infectious diseases, air pollution and ecosystem disruption (including access to fundamental rights such as water, sanitation, education and food) [6]. Numerous studies have found associations between higher temperatures and political violence and between natural disasters and armed conflicts [7]. Reasons for this include economic crisis and mass displacement, which exacerbate tensions.

Most deaths related to air pollution are anthropogenic, including the burning of fossil fuels; the pollutant responsible for most deaths is particulate matter (PM). Air pollution is categorised into indoor air pollution and ambient (outdoor) air pollution. Globally, air pollution has a devastating impact on child health, leading the WHO to declare it a public health emergency; 99% of children globally are breathing ambient air that is harmful to their health and above the WHO recommended limits [8]. In 2016, seven million deaths were linked to air pollution, and 9% of these were children [9]. Exposure to air pollution during pregnancy is associated with low birth weight and respiratory disorders later in life for children. Exposure in childhood is associated with increased hospital attendance, asthma, atopic dermatitis, allergic rhinitis, and allergic conjunctivitis [6,10].

Children have a right to a clean and safe environment, which includes clean air. Equally, they have a right to safe water and sanitation, free and quality education, and nutritious food. All of these determine the outcomes for health. Half of the increased child and maternal survival rates in recent decades can be attributed to increased access to these health determinants [11,12]. They are highlighted as fundamental in the African Charter on Human Rights, the United Nations Convention on the Rights of the Child (UNCRC) and the United Nations Convention on Economic, Social and Cultural Rights. These rights are also among the Sustainable Development Goals (SDGs). However, climate change is undermining child rights and impeding SDG progress [10]. Unequal access to fundamental rights, between and within countries, due to the current and historic asymmetric distribution of power and income means that climate change will have the most impact on the vulnerable, which are children in lower-income countries [13]. These countries, with the least fiscal space, will need to spend the most, relative to their resources, both to adapt to the changing climate and on loss and damage after extreme weather events.

## Cross-border responsibilities and duty bearers

This situation is unjust given that lower-income countries currently produce 14% of greenhouse gas emissions compared to the wealthier countries in the world. High- and upper-middle income (hereafter higher-income) countries produce 86% of greenhouse gases but are home to half of the world's population [14,15]. Varying responsibility for the climate crisis is recognised as a principle of international environmental law. It acknowledges that all states are obligated to address global environmental destruction, but not all are equally responsible; states have common but differentiated responsibilities [16]. This and the principle of equity are reflected in the Paris Agreement. The UNCRC recognises cross-border injustices and passed a historic and hugely important ruling in 2021 that 'a state party can be held responsible for the negative impact of its carbon emissions on children's rights both within and outside its territory' [17]. Indeed, a report by the Office of the United Nations High Commissioner for Human Rights concluded that to uphold the right to health, governments, civil society, the private sector, international partners, and individuals must work together to protect the environment [18]. This and the impact on children have been recognised in the Kyoto Protocol and the Paris Agreement, where 196 parties have committed to limit global warming to well below 2 degrees Celsius, ideally 1.5 degrees [16].

## Vulnerability to climate change: Colonialism and its sequalae

Much of today's global inequalities between countries can be traced to colonialism; many lower-income countries are former colonies, and many higher-income countries are former colonial powers. Consequences of colonialism include underdevelopment, unequal trade of resources, deculturalisation and deindustrialisation [19]. The purpose of colonial institutions was to extract economic returns for the home country, often neglecting and impeding the development of the host country. Sometimes, small cadres of economic and political elite were promoted, which caused inequality. This inequality may help explain why many former European colonies still have poor economic outcomes [20,21].

In the postcolonial era, many foreign powers continue to protect and project interests and preferences on old colonies. Many independent states are still controlled by foreign business enterprises, sometimes in collaboration with domestic elites [22]. One example of this is 'the resource curse', which is a counterintuitive phenomenon where countries with valuable natural resources, such as copper and iron, experience poorer economic growth than those with less mineral extraction. Some scholars argue it has led to a 'second scramble' over Africa, and if stronger institutions, development and better governance had been prioritised during the colonial era, perhaps the resource curse would be less prevalent today [23]. This is pertinent as many of the minerals required for renewable technologies are likely to be extracted from Latin America and Africa, and if the past trajectory is any indication, most minerals will be exported in raw form [24,25]. Foreign powers sometimes reserve former colonies as exclusive investment and trading locations under their sphere of influence, which is another example of imperialism [19].

## Vulnerability to climate change: Global governance

Several former colonial powers continue to dominate global governance and carry considerable voting power at the IMF and the World Bank. These global financial institutions have a massive influence on the policies and revenue in lower-income countries as these countries often require financial assistance, which often comes with conditions. Further, decision-making on international tax, which affects domestic revenue mobilisation, has been prescribed for the last 60 years by the Organisation for Economic Co-operation and Development (OECD).

Members include many former colonial powers, many of whom are most complicit in enabling corporate tax abuse [26]. This, in turn, hinders governments in raising revenue through tax to finance public services, to make progress on the SDGs, and to respond to the climate crisis [27].

No topic is currently dominating global health more than the climate emergency, yet global health institutions are most often located and run in higher-income countries, despite lower-income countries experiencing most of the impact. Higher-income countries hold the purse strings, house the most impactful academic journals, and award degrees which are often inaccessible to those in lower-income countries [28].

## Vulnerability to climate change: National governance

Despite the recognition of cross-border injustices, the risk that climate change poses to lower-income countries and the responsibilities of higher-income countries, there is no globally agreed approach to define a state's 'fair share' of the global burden of mitigating climate change [29]. This results in states avoiding responsibility for the collective outcome and hinders just global efforts to address the climate emergency.

In fact, six Portuguese young people are seeking a decision from the European Court of Human Rights requiring urgent action by 33 European governments. The purpose of the case is to prevent governments from avoiding responsibility and resolve the ambiguity around the fair share issue [30]. There is no agreed definition of what a fair share is, but the approach taken by the Climate Action Tracker is to rate country pledges on emissions and climate finance and assess the combined effect on global warming and consistency with the Paris Agreement if all governments were to put forward nationally determined contributions (NDCs) with the same level of ambition as pledges[31].

## Higher-income countries and global child health

Higher-income countries impact health in lower-income countries through multiple pathways; see Fig 1 [32]. While it is possible to quantify the impact of higher-income country policies on lower-income countries through aid, tax and trade, it is much more challenging when it comes to climate change. This is because of the collective responsibilities for the emergency. Nonetheless, given that the UNCRC has found that a state party can be held responsible for the negative impact of its carbon emissions on the rights of children outside its territory, we aim to review the role of one higher-income country (the UK) acting via the pathways in Fig 1

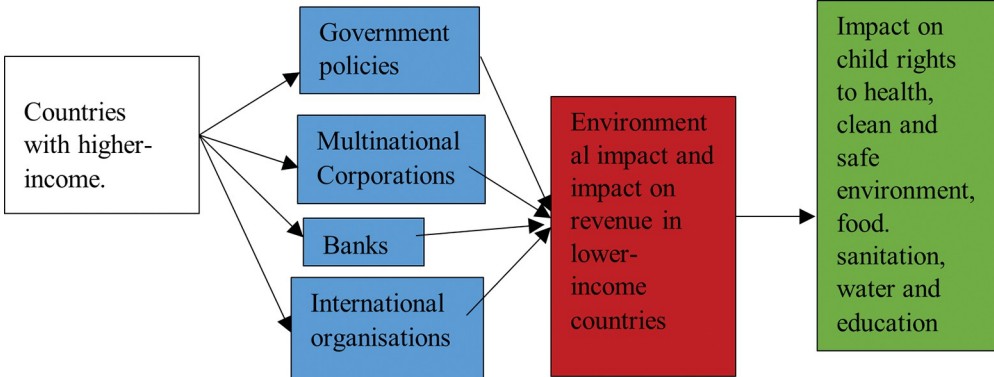

**Fig 1. The impacts of higher-income countries on lower-income countries act through different pathways [33].**

on climate change. To bring home the effects on children's rights, we shine the spotlight on one lower-income country (Malawi). Malawi has contributed little to the climate crisis, both historically and presently; in 2019, per capita, Malawi emitted 0.08 tonnes of carbon dioxide (CO2), on the other hand, per capita emissions in the UK was 5.48 tonnes [14].

To illustrate the pathways at play in Fig 1 we study Malawi, because it has strong historical links with the UK dating back over 150 years, from David Livingstone's expedition in 1859 to being subjugated to British rule from 1891. Now, both countries are members of the Commonwealth. In addition, Malawi receives aid from the UK and is one of Scotland's four partner countries. Malawian children are classified as high risk on UNICEF's Children's Climate Risk Index (CCRI). The CCRI considers both exposures to climate and environmental hazards, shocks, and stresses, and children's vulnerability to these. Children's vulnerability depends on their access to quality, sustainable services, such as clean water, sanitation, education, nutrition and health services [33,34]. The CCRI's objective is to determine the likelihood of deepening deprivation because of the climate emergency [34]. To reduce Malawian children's vulnerability and increase resilience, the government will require additional revenue to provide key services. To reduce Malawian children's exposure, those countries most responsible need to reduce global greenhouse gas emissions urgently.

The UK's medium-term policies that will impact the vulnerability of Malawian children to climate change include the quality and quantity of adaptation finance, which is channelled either directly from the UK or through international organisations. In addition, the UK's role in enabling revenue losses from Malawi has a critical impact on the Government of Malawi's resources.

The UK's long-term policies on mitigation consider the efficacy of policies to reduce global warming and includes the impact of entities the UK could regulate, such as multinational corporations (MNCs) and banks headquartered in the UK.

## Aim

The aims of this study are to assess the impact and the projected impact of climate change on child health in Malawi and to determine which UK policies directly impact Malawi's ability to adapt to climate change and how the UK could improve on climate change mitigation.

## Methodology

### Research questions

1. What impact is climate change having on child health in Malawi?

2. Are the international and UK organisations that receive finance from the UK government providing adequate and effective adaptation finance for Malawi? Is the UK paying their 'fair share' of climate finance? Is the UK taking adequate action to reduce their role in enabling revenue losses from Malawi?

3. Are UK policies and the practices of entities it regulates (including MNCs and banks headquartered in the UK) doing their 'fair share' and taking effective action to mitigate climate change and thus the impact on the health of children in Malawi?

A critical review of the academic and grey literature was used to answer the three research questions and conducted by the lead author. The search engines Knowledge Network, Web of Science and Google Scholar were used. The following search terms and combinations were used; 'Malawi' and or 'climate change', 'air pollution', 'climate compatible development', 'climate finance', 'Tobacco', 'Sugar'. 'Environmental impact of sugar', 'Environmental impact of

tobacco' 'Climate Change performance Index' 'Climate Change and Health', 'Effectiveness' and or 'climate finance' 'Green Climate Fund' 'Global Environment Facility' 'Nationally Determined Contributions'.

Relevance was assessed using the following inclusion criteria: the impact of climate change on child health in Malawi, the UK's climate finance policies, the climate finance policies of organisations that the UK contributes to financially, UK adaptation projects. UK policies on mitigation, including the policies and practices of banks and MNCs headquartered in the UK with operations in Malawi.

Online searches using Google were conducted to identify relevant grey literature, including reports, policy briefs, relevant MNCs (corporate social responsibility reports) and banks headquartered in the UK and working papers. The search was supplemented using publications of the government (UK and Malawi), the UN, and the WHO and citation searching of critical papers.

For climate change impact on child health in Malawi, Helldén et al's conceptual framework was used. To assess the UK mitigation strategies, the Climate Change Performance Index (CCPI) was used. It ranks countries who are responsible for 90% of global greenhouse gas emissions according to their climate protection policies. The methodology incorporates the most recent version of the Paris Agreement [35,36]. To address the question of 'fair share' for mitigation efforts and adaptation contributions the Climate Action Tracker was used [31].

## Results

The results are presented in three sections:

1. The impact of climate change on child health in Malawi

2. Adaptation finance and the UK's role in revenue loss

3. A review of the UK's contribution to the global burden of mitigating climate change (is the UK doing its fair share?)

### 1. Climate change and child health impacts on Malawi

In Malawi, the average temperature rose by 1 degree Celsius between 1901 and 2020, slightly less than the global average rise of 1.18 degrees Celsius. Since 1961, more inter-annual climacteric variations have been observed [37,38]. As a result, extreme weather events have been increasing, including droughts and floods, and the mean seasonal rainfall has been decreasing [39]. In 2019, 25,103 under-five children died in Malawi, mostly from preventable causes, such as pneumonia, malaria and diarrhoeal illness, birth asphyxia and trauma [40]. Pneumonia, malaria and diarrhoeal disease are exacerbated by climate change, with pneumonia linked to air pollution. Just over two-thirds (67.5%) of the population live on less than US$2.15 per day, which is defined by the World Bank as living in extreme poverty [41]. This makes Malawi more vulnerable to the impacts of climate change.

**1.1 Air pollution.** There are significant issues with indoor air pollution in Malawi due to low access to clean and safe cooking fuels; in 2016, only 2.5% of the population had access to clean fuels for cooking [42]. This fuel poverty means people use biomass fuels, such as wood, charcoal, and crop residues, for cooking, heating, and lighting. In 2019, 12,379 people died in Malawi because of indoor air pollution; 4,452 of them were children under five (17% of under-five deaths). Thus, indoor air pollution contributes to almost one-fifth of all deaths in children under five [42]. The effect on children's health is brought home in a cross-sectional study of 804 children to analyse the benefits of cleaner cooking methods. This study showed that 16.6%

of children reported chronic respiratory symptoms and one-fifth of the children had been admitted to hospital with respiratory symptoms. Lung function parameters were generally lower compared to the global average; most children had carboxyhaemoglobin levels above the WHO recommendation, and half had carbon monoxide levels above WHO recommendations. Children from households using safer cooking methods had lower carboxyhaemoglobin levels and better lung function [43].

Outdoor air pollution in Malawi caused 415 deaths (3%) in the under-five age group in 2019. This number is likely to rise as carbon dioxide emissions are increasing steadily, and air pollution increases in all countries with industrialisation before plateauing and decreasing again [44]. Approximately 1.4% of total mortality, 0.5% of all disability-adjusted life (DALYs) and 2% of all pulmonary diseases are attributable to outdoor pollution in Malawi [44].

**1.2 Change in vector patterns.**   Climate change creates more favourable conditions for disease transmission, including causes of diarrhoeal illness, such as *Vibrio cholera*, *cryptosporidium*, *Escherichia Coli* and rotaviruses. Diarrhoeal illness is the second commonest cause of under-five mortality worldwide and it is predicted an extra 48,000 children will die by 2030 if we continue our current trajectory. In Malawi, diarrhoeal illness accounted for 6% of under-five mortality in 2017, equating to 1,692 preventable deaths [45–47]. Other notable infectious diseases on the rise because of climate change include malaria, which accounted for 8% (2,257 deaths) of under-find mortality in Malawi in 2017 [48], dengue fever and *Borrelia burgdorferi* [6].

**1.3 Ecosystem disruption.**   Climate change threatens access to socioeconomic determinants of health, including water, sanitation, education and food [1]. For example, only 21% of the population has access to basic sanitation, and 54.8% have access to basic water [49]. Extreme weather can damage schools and hospitals as well as water and sanitation facilities; this reduces access to these fundamental rights, reduces climate change adaption capacity, and increases mortality. Economic modelling shows Malawi will lose the equivalent of 5% of its GDP because of climate change if global warming is limited to 2 degrees. If the global temperature rises as much as four degrees, Malawi will lose 13% of its GDP each year [50].

Droughts, changing precipitation patterns and high temperatures can cause crop failure increasing undernutrition, poverty and migration. This is particularly relevant to Malawi as agriculture is the backbone of its economy both at the household and national levels. Malawi's worst recorded famine in 2002 was caused by erratic rainfall and localised flooding and estimates suggest up to 3,000 people died [51]. Tobacco yields contribute almost 40% to the country's export earnings and are predicted to decrease by 45% because of climate change. Macadamia is a potential alternative, but areas suitable for cultivation are predicted to decrease by 21.6%. Furthermore, tea growing areas are predicted to decrease and maize yields, the most important crop for food security, are predicted to decrease by 50% [52,53].

Signs of undernutrition include being stunted, wasted and underweight. In 2020, 40.9% of children under five were stunted, 0.6% were wasted, and 9% were underweight [54]. Economic shocks due to climate change threaten to increase malnutrition [6]. Malnutrition leads to other health implications, including a weakened immune system, delayed growth, and altered brain development. Adults who were malnourished in childhood are more likely to experience insulin resistance, diabetes, hypertension and dyslipidaemia. In addition, undernutrition in early life is associated with lower-paid jobs and a reduced working capacity in manual jobs [55–57].

**1.4 Temperature changes.**   Malawi is predicted to experience more heat stress days (where atmospheric temperature exceeds 39 degrees Celsius; at these temperatures, health risks from heat are likely), and between 2070-2099, Malawi will likely experience 5-12 heat stress days per month for up to four consecutive months. The same study also predicted that the adaptive capacity in Malawi was low, highlighting the need for increased adaptation

finance [58]. For children, heat stress days increase the risk of heat stroke, electrolyte imbalance, kidney, respiratory and infectious diseases, and hospital attendance.

**1.5 Flooding and precipitation.**   Floods have increased in Malawi, and there have been 19 significant floods over the past five decades [59]. The most recent Cyclone Freddy, in early 2023, affected 883,000 households and 1.35 million people (over half were children), displacing 659,278 people, and killing at least 676 people. The recovery may take five years and cost more than US$400 million. Just four years earlier, Cyclone Idai in March 2019 affected 975,000 people, with 86,976 displaced, 60 killed, and 672 injured; the estimated recovery cost from this flood is US$370.5 million, this is 18% of annual government revenue [49]. Governments from lower-income countries will increasingly have to mobilise funds for disaster recovery, diverting resources from development and access to fundamental rights. Floods negatively impact access to fundamental determinants of health by destroying schools, sanitation facilities and hospitals [60]. This flood saw a rise in infectious diseases including diarrhoea, cholera, malaria and coughs [61]. Another major flood in January 2015 affected 638, 000 people [62]. Children are more at risk of drowning and injuries during flooding and suffer more severely from the infectious diseases mentioned [6,10,63]. These extreme weather events also have a negative impact on child mental health [6]. The changing precipitation pattern threatens Malawi's agriculture-based economy and further increases food insecurity [64].

**1.6 Droughts and wildfires.**   Malawi has experienced seven droughts over the last five decades that are increasing in frequency and magnitude [61]. Droughts increase the spread of infectious diseases and affect crop growth and hence childhood nutrition. In more extreme circumstances, they increase poverty and drive migration [6]. Since the early 2000s, 128 countries have seen an increase in wildfires; this can adversely affect children through burns and harm from smoke inhalation. In addition, wildfires damage vital infrastructure, such as schools, hospitals, sanitation and water systems, and cause displacement of communities [6,60]. Malawi is experiencing more days with favourable conditions for wildfires [64].

**1.7 Summary.**   Climate change is impacting child health in Malawi in multiple interlinked ways as discussed above. The health of the planet and human health are inextricably linked therefore to minimise harm to child health in Malawi, development must be green. For example, energy sources must come from clean renewables, which will reduce the impact of air pollution. With development will come increased access to core socioeconomic rights, such as clean water, sanitation and education, which will allow Malawians to adapt better to the health impacts of climate change. However, to do this the Malawian government needs fiscal space, and repairing loss and damage because of climate change is draining resources. Higher-income countries, such as the UK, can help with this through climate finance and ensuring they do not facilitate government revenue leaks. Finally, given that higher-income countries are driving the climate crisis, they must do everything they can to mitigate against climate change. We explore these issues below.

## 2. Adaptation, climate finance and revenue losses

**2.1 Overview of Climate finance and 'Fairness'.**   The UK has signed multiple international treaties calling for parties with more financial resources to assist those who have less and are vulnerable. Treaties include the United Nations Framework Convention on Climate Change (UNFCCC), the Kyoto Protocol, the Paris Agreement, and the Cancun Agreement [65]. The Cancun Agreement (2010) pledged to mobilise US$100 billion to support developing countries, which are mostly Non-Annex 1 countries under the UN Framework Convention on Climate Change and most vulnerable to the adverse impacts of climate change

An analysis by the OECD in 2017 estimated US$71 billion of climate finance was being spent in developing countries and this has been increasing since 2013. However, less than ten

percent of global spending on climate finance goes to developing countries. Of the US$71 billion transferred in 2017, 19% was spent on adaptation, 73% on mitigation and 8% on projects that covered both [66,67]. Mitigation projects are usually in the form of loans, and most adaptation projects are in the form of grants. Almost three-quarters of the climate finance, going to developing countries, is in the form of loans and 14.5% of these loans were from private sources, just over a quarter was in the form of grants [68,69]. Loans can increase the risk of debt distress if irresponsibly structured [70]. As well as poor quality, the amounts are paltry when compared to the need; the United Nations Intergovernmental Panel on Climate Change reports that to limit global temperatures rises to less than 1.5 degrees Celsius, an extra US$830 billion annual spending for the years 2016 to 2050 will be needed [71]. For adaptation in Africa alone, US$50 billion per year to 2050 is needed; currently, the continent gets between US$1 and US$2 billion [72].

Concerns about climate finance from developed countries include transparency and accountability in recipient countries. Criticisms include how funds are managed and predictability and adequacy of funds, for instance, the USA withdrew their funding for the Green Climate Fund (GCF) in 2019. Some have expressed concerns about fairness and equity in the decision-making processes. This is supported by a recent report that found that the most vulnerable countries are least likely to receive adaptation finance from either bilateral or multilateral donors [73].

**2.2 The UK's contribution to climate finance.**   Between 2016 and 2021, the UK contributed GBP5.8 billion to international climate finance, 47% went to adaptation projects and the rest was spent on mitigation [74]. This contribution was classed as Official Development Assistance (ODA), but climate finance should be in addition to ODA, this is breaking a UN-brokered agreement [16,31]. The UK's contribution increased to GBP11.6 billion for the period of 2021 to 2026, but it was also taken out of the ODA budget [31].

Assessing the impact of climate finance paid by the UK to Malawi is further challenged because the UK provides climate finance either directly (bilateral funding) or via international organisations (multilateral funding) including the GCF, Global Environment Facility (GEF), the Nationally Determined Contribution Partnership (NDC Partnership) and the Climate Investment Fund (CIF) [74–77]. Some projects also span more than one country.

The Climate Action Tracker gauges whether countries are paying their 'fair share' of climate finance based on transparency and adequacy of the support. Another study assessed fair share based on carbon emissions since 1990 and gross domestic income. The findings are that the UK contributes 48% of its 'fair share' to climate finance. This is compared to Norway, which contributes 188% and at the other end of the spectrum, the US which contributes 4% of its 'fair share' [78]. Based on an inadequate amount, decreasing climate finance compared to 2015 and climate finance being counted as ODA, the Climate Action Tracker rated the UK's contribution to climate finance as 'highly insufficient'. It also highlights that, for the UK to meet their 'fair share' of mitigation for the climate crisis and meet the Paris Agreement they need to support climate action in other countries. The Scottish government has also agreed to treble its budget to GBP36 million for 2021-2026, but they are not subject to the same international treaties as the UK government.

**2.3 The UK's contribution to climate finance in Malawi.**   Most of the climate finance from the UK to Malawi is channelled through international organisations, including multilateral development banks. Benefits include enhanced stakeholder engagement, reduced duplication, country ownership and increased sharing of expertise. Some, such as the CIF, can also draw on further co-financing opportunities and the private sector. However, this can increase the risk of debt distress. International climate finance organisations need to determine what counts as a fair contribution to climate finance and need to address transparency [74,76,79–83].

**2.4 Effectiveness of projects in Malawi that the UK has supported with climate finance.** Research assessing the effectiveness of climate finance has been scarce in Africa, indeed there is no agreed definition of effectiveness [84]. This is further complicated by the complexity of climate finance.

The UK has supported over 70 different projects in Malawi; the vast majority of these are funded through combined grants and co-finance, 22% of projects were through grants only and 1% of projects were co-finance only [74]. The projects that have been assessed for effectiveness are summarised in Table 1 [74,76,85]. Assessing effectiveness of individual projects that the UK supports is challenging given the complexity of climate finance, see Table 2 [86–91].

One study examined the effectiveness of two projects in Malawi: the Developing Innovative Solutions with Communities to Overcome Vulnerability through Enhanced Resilience project (DISCOVER) (not UK financed), and the Enhancing Community Resilience Programme (ECRP) which the UK helps to finance. A minority of households experienced benefits, including increased wealth, improved food security, and better nutrition. The adaptation benefits were modest and include reduced vulnerability to dry spells due to improved soil quality and moisture, as well as better housing and farmland for protection against heavy rain floods and winds. Other adaptation benefits include the ability to grow food year-round and access to emergency finance in the event of climate shocks. The adaptation benefits were still sensitive to extreme weather events which may have led to misperception and under-reporting of benefits. On the other hand, participants may have been reluctant to give negative feedback [86].

**Table 1. UK climate projects in Malawi.**

| Project | Assessed Benefits of Project | Areas for Improvement |
|---|---|---|
| ECRP<br>Objective: To improve access to services, strengthen flood resilience, and enhance institutional capacity for local service delivery and integrated disaster risk management at the national and sub-national levels [86]. | Increased wealth<br>Improved food security with all-year/dry season agriculture<br>Better nutrition,<br>Reduced vulnerability to dry spells due to improved soil quality and moisture<br>Better housing and farmland for protection against heavy rain floods and winds<br>Access to emergency finance when climate shocks occur<br>Benefits spreading to non-targeted areas | Adaptation benefits still very sensitive to extreme weather events.<br>Early withdrawal of financial support<br>Some households lost money<br>Projects unrealistic to some situations, e.g., households need initial cash to buy equipment so poorer households and female headed households less able to take part<br>Difficult to get cash crops to market<br>Increase in inequality in villages led to theft |
| Solar PV projects rural Malawi [87] | Provision of electricity to rural public facilities | Low sustainability due to low local community stakeholder engagement<br>Lack of key management positions<br>Limited training provisions<br>Affordability<br>Not meeting consumer demand due to battery and panel size |
| World Agroforestry Centre which has helped agroforestry and conservation agriculture [88] | Improved soil quality, reducing the need for fertiliser, as well as increased crop yields, reduced weeds and improved water filtration | |
| Malawi farming systems [89] | Improved natural resource management and forests in better condition | Sustainable scaling up of projects is needed |
| Malawi fishing industry [90,91] | Improved marketing and fish processing, breeding fish better adapted to hotter climates | These adaptation techniques are still limited, and livelihood diversification is important |

**Table 2. Climate finance projects in Malawi supported by the UK government.**

| Fund | Given by UK | Received by Malawi | Projects | Date of project and type of finance |
|------|-------------|--------------------|----------|-------------------------------------|
| Green Climate Fund (GCF) | No specified amount | US$35.3 million | Climate friendly cooling technologies and systems | 2021 - no end date set, grant, loan and co-financing |
| | | | Renewable energy | 2018-2037, grant and co-financing |
| | | | Enhanced early warning weather and climate information systems | 2015-2023, grant and co-financing |
| Climate Investment Fund (CIF) | US$2 billion | US$0.76 million | Pilot programme for climate resilience, including water security | 2020 - no end date set, grant, co-financing, loan |
| Global Environment Facility (GEF) | GBP250 million | US$137 million | Invest in nature, supports international conventions on key topics, such as biodiversity, chemicals, climate change and desertification | Projects are mostly funded by grants and co-financing. 11/66 projects (past and present) listed on the Global Environment Facility database are grants only. |
| Nationally Determined Commitments partnership | No specified amount | No specified amount | Solar panels and heating | |
| Scotland's climate justice fund | GBP36 million | No specified amount | Adaptation to find alternative sources of income, such as livestock rearing, fish farming, honey production; improve water irrigation; ensure safe drinking water and water resiliency | |

There is evidence these benefits are spreading to non-targeted areas and helped with development, for example, increased forestry activity and access to firewood, but time is needed to assess the full mitigation benefits and shortfalls. These projects also benefited local governments, non-governmental organisations and donor agency employees, who reported improved reputation, cohesion, capacities, innovativeness, access to resources, and lobbying influence [86].

Shortfalls of these projects included early withdrawal of financial support, threatening longevity, and the extreme weather events experienced during the projects still hindered their success. Some households reported they lost money because of participation in the projects and found the projects unrealistic for their situation. For instance, cash crops were difficult to get to market. Increased inequality in villages and perceived wealth also led to increased theft. Female headed households and resource poor households benefited the least from the projects; reasons for this included the requirement for initial investment to kick start the project, which these households could not spare [86].

Other studies found low sustainability results for solar PV projects in rural facilities in Malawi [87]. The UK contributes to the World Agroforestry Centre which has helped pioneer agroforestry and conservation agriculture in Malawi, including integrating trees between crops and focussing on soil and water management. This has improved soil quality, reducing the need for fertiliser, as well as increased crop yields, reduced weeds and improved water filtration [88]. It also increases biodiversity and helps with carbon capture. Other projects have enhanced natural resource management leading to forests that are in better condition and better managed [89]. The fishing industry in Malawi has adapted through marketing and improving fish processing strategies, and there are projects looking at breeding fish more adapted to living in hotter climates, however, these strategies have their limitations and it is important to diversify livelihoods as a method of adaptation [90].

**2.5 The UK and revenue losses in Malawi.** The UK and dependencies create about one-third of the vulnerabilities to tax abuse globally. Malawi loses about 3.16% of government

revenue each year to tax abuse, this amount could allow 12,000 and 20,000 people to have access to basic water and sanitation, respectively, each year, and an additional 5000 children would attend school every year [27]. By increasing access to these rights, Malawian children would be better adapted to deal with the consequences of climate change [27,34]. Further, Malawian taxing rights are potentially undermined by a double tax treaty, called the Income Tax Treaty, which entered into force between the UK and Malawi in 1956, before Malawi gained independence. Many treaties signed in that period aimed to eliminate double taxation in favour of capital exporting countries, which tended to be richer nations, in this case, the UK [92]. There have been efforts to redress this globally, such as through the development of a UN model tax treaty, which gives greater taxing rights to source countries, like Malawi, in contrast to the OECD's model agreement. However, even the UN model tax treaty is not without its faults from a lower-income country perspective [93]. The UK-Malawi tax treaty has not been amended to reflect the UN model tax treaty. Double tax treaties limit the reach of domestic legislation by reducing tax rates, and in the UK-Malawi tax treaty, for example, there are very low withholding tax rates, much lower than Malawi's domestic tax rate. Tax treaties may also be abused for tax-planning schemes, in the form of treaty shopping, which puts Malawi's potential revenue at risk [94]. Fiscal deficits, because of tax abuse, drive debt accumulation and debt servicing, which diverts resources from climate change adaptation [70]. In 2023, Malawi diverted almost 25% of revenue to service debt.

The UK does not require all MNCs with headquarters within its borders to publish country-by-country reports although some companies choose to as part of standards from the voluntary Global Reporting Initiative (GRI) [95]. At present, only MNCs in the extractive industries and banking sector are required to publish country-by-country reports, but these are not comprehensive [96]. The Global Report Initiative's (GRI) tax reporting standard for public country-by-country reports includes information on every tax jurisdiction where the multinational group does business. It includes information on revenue, pre-tax profits, taxes paid, such as corporate income tax, and other relevant financial and business information, such as number of employees. Typically, MNCs report only at the group level. The opacity means it is very difficult for tax authorities to assess the practices of subsidiaries operating within their jurisdiction and to enforce tax regulations – and this is particularly important for countries like Malawi that rely on corporate income tax. Transparency in tax reporting also helps other investors and law makers in improving international and domestic tax rules and creates a fairer playing field for domestic companies. The Tax Justice Network and Tax Justice UK suggest that public country-by-country reporting in the UK would also help domestically to increase the UK's corporate income tax by GBP2.5 billion per year [97]. Some companies opt to voluntarily publish country-by-country reports, or payment to government reports, but the tobacco and sugar companies and service industry operating in Malawi with headquarters in the UK do not report voluntarily. Unilever does tax reporting on a country-by-country basis and has a policy on fair tax but it does not appear to report on pre-tax profits, revenue and tangible assets on a country-by-country basis [98]. Pearson plc are GRI compliant, they have a tax policy and have included a full breakdown of revenues, profits and tax liabilities on a country-by-country basis [99].

Banks in the Isle of Man, a UK crown dependency and known tax haven, held $11 million of Malawian capital and other banks in the UK also held US$11 million as of the third quarter in 2022 [100]. Gains from capital are less likely to be taxed if they are held in tax havens, thus depriving the Malawian government of tax revenue. In most countries, it is the wealthy who hold their wealth offshore. This reduces resources for public services, which drives inequality and increases the tax burden on those least able to afford it.

## 3. The impact of UK climate policy on mitigation and children in Malawi

The Climate Action Tracker assessed the UK's NDCs as 'insufficient'. This means the UK is not putting in their 'fair share' of the effort to meet the Paris Agreement. If all countries were to follow the UK's approach, global warming could reach almost 3 degrees Celsius [31].

For the CCPI ranking in 2022, the UK was ranked seventh, scoring 73.09 (of 100). No country achieved the best performance rating. The UK ranked highly for reduced energy consumption, greenhouse gas emissions and climate policy, however, they were ranked medium for their renewable energy use. Critically, the index highlights that the UK has the political and financial support to reach net zero by 2050 and has created promising policies to support the development of key technologies, such as electric cars and carbon capture. The CCPI report highlights that in the UK, carbon credits and fossil fuel subsidies are problematic and recommends the urgent use of farm subsidies to restore biodiversity and land use as a carbon sequester.

The crucial areas for improvement in UK climate policy are inclusion of overseas emissions, wider employment of renewables, improved land use and low-carbon heating. In 2016, heating was the largest contributor to the UK's carbon footprint, contributing 9.7% [101].

**3.1 Overseas emissions.** The UK is one of the 13 countries that has committed to reach net zero carbon emissions by 2050 and UK greenhouse gas emissions are decreasing [14]. Between 1990 and 2016 domestically, emissions decreased by 41%. However, 46% of the UK carbon footprint occurs overseas and this has increased by 14% since the 1990s. The UK's domestic climate change policy, which aims to reach net zero by 2050, does not account for overseas emissions, although they have now included international aviation emissions in their NDCs, but these are not legally binding. The true reduction including the overseas carbon footprint in UK greenhouse emissions was 15% between 1990 and 2016 [31,102].

**3.2 Renewables, carbon pricing and Research and Development.** The UK has no carbon tax and no cap-and-trade system (where companies have a cap on emissions but can buy or sell emission allowances), however, it does have extensive excise taxes on oil and fuel. Fossil fuel subsidies in the UK are contentious. The UK government states that it does not give subsidies and follows the approach of the International Energy Agency, which defines subsidies as taking measures to reduce fossil fuel prices below world market prices. However, the UK government does give tax breaks to companies for fossil fuel exploration as well as research and development, which many would argue are a subsidy. Some sources calculate that the UK spent US$186 per capita in fossil fuel subsidies in 2020; this has decreased from US$306 in 2015 [103,104]. In the UK in 2020, 78% of energy was still supplied by fossil fuels, including 3% from coal [105]. Although there are clean energy alternatives, the UK still needs storage solutions and green alternatives to vital commodities such as cement and steel [106]. Thus, the UK needs to increase its research and development spending on greenhouse gas neutral replacements, but this has decreased since 1990 [107].

**3.3 Land use.** The UK Climate Change Committee concluded in 2020 that 20% of agricultural land will need to be released by 2050 for actions that reduce emissions and sequester carbon [108]. It highlights five key areas that the UK needs to focus on; these include reducing consumption of the most carbon intensive foods, bioenergy crops, restoring peatlands, increasing UK forest cover from 13% to at least 17%, and introducing low-carbon farming practices.

**3.4 Multinational corporations working in Malawi with headquarters in the UK.** Determining which MNCs were relevant was a challenge; the UK's company register, Companies House, does not provide a full list of companies registered in the UK. To obtain a list Wikipedia was used and some companies may have been missed. This lack of transparency is problematic and highlights fundamental issues with UK tax structures as discussed in section 2.3.

To assess the impact of MNCs, their carbon emission reporting was reviewed. Carbon emission reporting covers three scopes. Reporting on scope one and two emissions are mandatory under UK law, scope three is also mandatory but only for certain criteria. Companies are now required to report on a subset of five scope three emissions out of fifteen [109]. Scope one covers direct emissions, scope two covers indirect emissions from purchased electricity, heat or steam, and scope three covers indirect emissions not included in scope two, for example those that occur in the supply chain. Scope three reduction has potential to have the biggest impact [110].

**3.5 The tobacco industry Malawi.**   Four tobacco companies operate in Malawi, and two of these are MNCs, Imperial Brands and British American Tobacco, with headquarters in the UK. Tobacco has harmful effects on the environment across the entire cultivation cycle; at the agricultural level, these harms come via land, water and pesticide use. Greenhouse gases are required during the transport and manufacture of tobacco and harmful waste is produced during the consumption and post-consumption process [111]. It also causes deforestation both through land clearance and the curing process.

Climate change in Malawi is exacerbated by the devastating rate of deforestation and biodiversity loss, between 30,000 and 40,000 hectares of land were deforested per year between 1975 when the forest cover was 47% and 2005 when it was 36%. This is the highest rate of deforestation in the Southern African Development Community (SADC) region [112].

Tobacco farming is the leading cause and was linked to 26% of deforestation in the early 2000s. In addition to this, the curing process requires wood; if air-cured, it needs wood to build the barns, and the fuel-cured tobacco requires firewood. However, it provides 67% of export earnings and farmers typically earn more through tobacco farming than other crops, such as maize, groundnuts and soybeans. Additionally, some tobacco companies provide loans, expertise and transport of the crop to market. Whilst this has benefits, there is growing concern that tobacco growing increases food insecurity due to many subsistence farmers foregoing food staples like maize. Tobacco farmers, despite earning more from tobacco, are usually poorer compared to their non-tobacco growing counterparts, due to the labour intensiveness of tobacco, costs of fuel and wood, and land rent. It is believed farmers in Malawi are intentionally cheated at market by systematic under-grading and hence under-pricing of their tobacco leaf [111]. Farmers also have to spend a larger proportion of their income on healthcare due to health harms from the crop [111].

Health harms include green tobacco sickness, a form of nicotine poisoning farmers can get while tending the crop. This can be avoided by tending the crop when it is dry or wearing the appropriate personal protective equipment. Sadly, children do a large share of tobacco growing and are among those who develop green tobacco sickness. As tobacco is a monocrop, it requires increased pesticide use and ranks number one for pesticide usage in Malawi, many of these are harmful and banned in some countries, however, farmers are using them with little personal protective equipment [111,113] and there are environmental residues from the pesticides [114]. Pesticides were reported as the leading cause of poisoning in Queen Elizabeth hospital in Malawi, and 79% of self-poisoning cases involved pesticides [115]. Pesticides have also been found in drinking water, although poisoning by this means is rare and there are no studies in Malawi reviewing poisoning from food ingestion. There are laws regulating pesticide use in Malawi, but there needs to be greater enforcement of these, greater education around pesticide harms, and mandatory reporting of poisonings. Greater financing and more staff at relevant institutions is also required to strengthen control, assess harm, and implement guidelines [115].

The most polluting part in the tobacco cycle is the manufacturing part; estimates from 1995 predict annually the industry produces 300,000 metric tonnes of nicotine contaminated waste and 200,000 tonnes of chemical waste. The carbon dioxide produced globally by the industry was estimated at 8.76 million metric tonnes in 2016.

The consumption of tobacco, whilst directly harmful to the consumer, has additional environmental harms. In a single year, tobacco smoke releases thousands of metric tonnes of known human carcinogens into the atmosphere as well as three greenhouse gases, carbon dioxide, methane and nitrous oxides. Plastic waste from packaging is a concern and post-consumer waste produces between 340 and 680 million kilogrammes of waste and contains 7,000 toxic chemicals. These toxic chemicals include human carcinogens, heavy metals and arsenic; research into effects on aquatic life and the wider environment is still ongoing [113].

Imperial Brands report on scope three emissions but not all fifteen; they highlight their work with partners to understand these better [116]. British American Tobacco reports on five of the fifteen scope three emission criteria and highlight this is 90% of their scope three emissions. They do not report plans to scale up their scope three reporting [117]. Environmental reporting is by no means comprehensive for either MNC [104]. For example, in their 2022 Environmental, Social and Governance (ESG) report, Imperial Brands state they are committed to making sure wood used in the curing process is sustainable but do not state how they will assess the sustainability of the wood or ensure the curing process does not contribute to deforestation [116]. British American Tobacco on the other hand talks about avoiding 'net deforestation'; they do not define this or appear to consider that natural forests and ecosystems are rich with biodiversity which takes centuries to develop but minutes to destroy [117]. Tobacco companies have attempted to offset deforestation in Malawi by distributing tree seedlings. However, many of these are distributed towards the end of the rainy season, resulting in poor survival, and these efforts are also poorly monitored by the tobacco companies. In addition, there is poor collaboration between the tobacco companies and governmental departments in Malawi [118].

In Malawi, it is prohibited to extract forest wood for tobacco processing, which 97% of farmers were unaware of this rule, at least according to one study[112]. In the UK, it is 'illegal for UK businesses to use key commodities if they have not been produced in line with local laws protecting forests and other natural ecosystems' [119]. A law passed in November 2021 'prohibits larger UK businesses from using commodities associated with wide-scale deforestation' [120]. Furthermore, member states of the WHO have adopted the WHO Framework Convention on Tobacco Control and agree to protect the environment and the health of persons in relation to tobacco cultivation and manufacture [113]. Malawi has neither ratified nor acceded to this treaty; however, the UK has.

**3.6 Sugar in Malawi.**   Sugar is the second largest export in Malawi, accounting for 9.5% of exports; and 97% of what is produced is exported [121]. Associated British Foods, an MNC with headquarters in the UK, sources some sugar from Malawi and states that 82% of its total energy use comes from the sugar division [122]. Sugarcane cultivation has likely caused more loss of natural habitat and biodiversity than any other crop on the planet; this is confounding the climate change crisis [123,124]. In addition, both sugarcane and sugar beet contribute to water scarcity; 1 kilogramme of sugar from cane and beet requires 1,500 and 935 litres of water respectively [124]. This is concerning for Malawi, where climate change is causing drought and extreme weather events threaten safe water supplies. Another area of concern is air pollution caused by the burning of sugarcane before cultivation, and the harms have been discussed in the air pollution section [125]. Other impacts include soil erosion, heavy use of agro-chemicals, discharge and run-off from polluted effluent, and pesticide use [123]. As mentioned in the tobacco section, some of these chemicals are banned by the European Union and sugar ranks third for pesticide usage in Malawi. The transport and processing of sugarcane can produce between 210g $CO_2$eq/kg to 630g $CO_2$eq/kg [124].

Some argue that sugar crop cultivation has positive environmental impacts, for instance, sugar beet in the UK increases nutrients in the soil and reduces soil erosion. Its use as a break crop also reduces pesticide and fertiliser use and sugar cane is an efficient biomass source [124].

The World Wide Fund for Nature (WWF) has recommended best management practices to reduce the environmental impact of sugar without profit loss. Considerations include use of waste to make other useful products such as paper, alcohol and fuel. Green harvesting techniques are an option, as well as integrated pest management as an alternative to pesticides. Based on Associate British Food's annual report, they are piloting a more efficient water irrigation system in Malawi, and they also make co-products such as animal feeds bioethanol and electricity. They state that they use more than 58% renewable energy; however, it is not clear if they have adopted the best management practices recommended by WWF. Their scope three reporting does not cover the fifteen categories and only reports on one of the business subsidiaries they own, Primark. They state they want to reduce their water footprint by 30% but still have not published data on this although, they include this is in their future plans [126].

**3.7 Unilever.**   Unilever is a consumer goods company with branches in Malawi; it has comprehensive goals for reaching net zero by 2039 and has made clear steps to achieve these, such as using renewable energy and even looking at laundry detergents that capture industrial carbon. They have picked key scope three targets which are comprehensive, but they fail to report on all of the fifteen criteria [127]. Further probing from Ethical Consumer has found Unilever still uses a high percentage of unsustainable palm oil, but it discloses all the relevant information and has targets in place [128].

**3.8 Service industry.**   G4S is a security business that has operations in Malawi; based on their website, they disclose their emissions to the carbon disclosure project, including scope three emissions, and have made commitments to reach net zero in by 2050. However, they fail to report on all fifteen of the scope three emissions criteria. Pearson plc is an education and publishing company that owns subsidiaries in Malawi; the website states that it aims to be carbon neutral by 2030. It also aims to have 100% of paper products certified by the Forest Stewardship Council (FSC) by 2025 and to increase their recycling and design digital products for energy efficiency. Its carbon footprint is published in the ESG report including scope three emissions and they appear to have comprehensive plans to reduce emissions. The company reports on all fifteen of scope three categories except eight, ten and fifteen which were not deemed relevant [129]. The company rated poorly on the Forest 500 ranking scoring 4/10; this ranking assesses commitments and actions of a company to ensure their activity does not further contribute to deforestation. It should also be noted that Pearson plc has signed up to the New York Declaration on Forests and is an Ethical Consumer Goods Forum member which provides information on ethical conducts of companies [130].

**3.9 UK banks.**   A recent report shows the world's top 60 banks, including UK banks, are still heavily invested in fossil fuels. Between 2016 and 2021 a total of US$4.6 trillion was invested and, despite the Paris Agreement, this has been increasing [131]. Of the 60 banks analysed, 44 have now committed to net zero by 2050 but these banks were still invested in companies expanding in fossil fuels. The only UK bank that has set a target to reach net zero by 2050 is Barclays, it is also the bank most heavily invested in fossil fuels in the UK. Other UK banks that have pledged to reach net zero by 2050, but lack targets, include the Lloyds Banking Group, HSBC, Ecology Building Society, Natwest and Standard Chartered [131,132]. UK banks will be contributing to climate change through their scope 1, 2 and 3 carbon emissions, but reviewing each UK bank with regards to this is beyond the scope of this article.

## Discussion

### Principal findings

This paper finds climate change is harming child health in Malawi and these harms are going to increase. As a result, children in Malawi can expect more food insecurity and threatened

access to clean water, sanitation, and education. Climate change threatens Malawi's agriculture based economy, thus further compounding access to these rights and increasing mortality rates. Children in Malawi will be more exposed to heat stress days, droughts and wildfires, floods and changing precipitation patterns. These not only threaten access to fundamental rights, but also come with their own health risks. Children in Malawi will be more exposed to air pollution, which already causes 17% of preventable deaths. In addition, they will be more exposed to life-threatening diseases such as malaria and diarrhoeal illnesses. Malawian children are at high risk of deepening deprivation and humanitarian crises because of climate change and their already vulnerable situation [34].

Whilst some progress has been made in tackling the crisis, these harms are being exacerbated by high-income countries, such as the UK, which are not doing their 'fair share' to support lower-income countries like Malawi to adapt to climate change. In addition, they are not doing their 'fair share' to mitigate against the climate crisis.

## Adaptation – climate finance and revenue losses

Despite these injustices, less than 10% of global spending on climate finance goes to lower-income countries like Malawi and the amount is vastly insufficient. Africa as a continent needs US$50 billion per year from now until 2050 in adaptation finance, yet it only receives US$1 to US$2 billion. The UK only contributes 48% of its 'fair share' to climate finance and less than 50% goes to adaptation projects. This should be in addition to ODA but is currently counted towards the UK's ODA. This contravenes the multiple international treaties which the UK is party to. Climate change and its associated health risks are already occurring; adapting to these to reduce harm is crucial. Mitigation for climate change will take decades to have an impact and whilst vital, climate finance in lower-income countries should give preference to adaptation. First, because they are the most vulnerable to climate associated shocks and they are not the drivers of climate change, climate compatible development that aids adaptation must be the focus. Second, adaptation is what many lower-income countries, particularly in Africa, have decided as their nationally determined contributions with regards to climate change [133]. Although most forms of adaptation finance are in the form of grants, most mitigation projects are in the form of loans, and debt servicing diverts resources from sustainable development and industrialisation, making countries less resilient to climate change.

The UK has recently doubled its contribution to climate finance, but urgently needs to review the balance between adaptation and mitigation. The UK should commit to significant contributions to the loss and damage fund which was agreed at the annual UN meeting on climate change, COP27. This is a fund that provides support to vulnerable countries to enable them to recover from the damage already caused by climate change [134]. Scotland has pledged to do this; the funds are coming from the Climate Justice Fund, they are not additional, and Malawi has already benefited from this fund [75,77]. As a key contributor to international organisations, such as the GEF, GCF and NDC partnership, the UK should advocate for increased reliable funding to meet the Cancun Agreement, UNFCCC, the Kyoto Protocol and the Paris Agreement. Equally, it should promote a fair and equitable process for allocation of funding, ensuring the most vulnerable countries receive their share of climate finance. Currently, other factors are ranked above vulnerability, possibly because the most vulnerable countries have the least developed markets, the least fund management experience and ability to develop bankable projects [135]. Multilateral donors tend to prioritise 'well-governed' countries, and bilateral donors often value previous bilateral relationships and strategic importance before vulnerability.

Assessing climate change adaptation projects, especially transnational ones, is challenging. One paper examining such a task made five recommendations. First, orchestration is key, and

projects need a bottom-up approach as well as a top-down approach. For example, providing farmers with adequate information. Second, effectiveness requires good process management. Third, the type of project affects where initiatives put emphasis on effectiveness; for instance, service providing, and standard-setting initiatives put more emphasis on outcomes which have stronger behavioural change. Fourth, high levels of institutionalisation made projects more effective, for example, having targets and external monitoring systems. Finally, effective co-ordination, partnership and working with other complementary projects improves effectiveness [136,137].

## The UK's role in tax abuse

The measures required to tackle tax abuse include the automatic exchange of information, beneficial ownership registration and country-by-country reporting [26]. Automatic exchange of information means countries automatically share relevant financial information on corporations and individuals; this makes it easier for other jurisdictions to trace illicit finance. Beneficial ownership is the practice of registering the person or people who own a company. Country-by-country reporting refers to the practice of publishing profits made and tax paid in each country where the corporation operates [26]. The UK should reduce loopholes within each of these categories to increase the chance of Malawi having adequate revenue to adapt to climate change. The UK-Malawi tax treaty should be reviewed and updated to give Malawi greater taxing rights, and the UK should support calls for a body and negotiations on a framework convention at the United Nations to govern cross-border tax transparency [138].

## Research and development

The UK also needs to increase research on climate-related harms in Africa as some of the world's most vulnerable populations reside there [139] with a focus on industrialisation and development pathways to support countries in the long-term, including improving resilience to shocks. Additionally, many African economies are heavily reliant on agriculture, which is sensitive to climate change. Despite this, less than 10% of papers that were published in 2017 relating to health and climate change focused on Africa specifically [139].

Equally important, the UK needs to increase research and development funding into climate change solutions, this sends a powerful signal to markets and can drive change. If the world was to use all the current available tools to reduce greenhouse gas emissions, there would be a 65% to 75% reduction relative to our current pathway; however, this would not bring us to net zero [140]. Innovations particularly important in the UK are greenhouse gas neutral heating and plant-based diets. Innovations relevant to Malawi include drought and flood tolerant food crops, agroecological approaches to farming, zero-carbon fertiliser and green cooking fuel alternatives [141]. Currently, the safe cooking fuel alternative in Malawi would be a fossil fuel or biogas, [42] which whilst reducing indoor air pollution would confound the climate change crisis. Other safe alternatives include electricity or solar cooking stoves, but these may not be available for all Malawian households due to cost and electricity access [142]. There is no consensus in the literature about the necessary increase in government spending on research and development spending; however, some entrepreneurs have recommended a five-fold increase over the next decade [141]. They also recommend that any innovations should be affordable for middle-income countries. It is futile for high-income countries to reach net zero if their emissions will be replaced by middle-income countries. The Climate Action Tracker highlighted that cooperating with other countries is key for the UK to meet its 'fair share' in climate change mitigation.

## Mitigation – the UK's climate change policy

The UK is leading the way in tackling climate change and could reach net zero by 2050; however, its climate change policy is not ambitious enough and it is not doing it's 'fair share' to mitigate against the crisis. To adhere to the Paris Agreement and ensure global warming remains below 2 degrees Celsius with a 95% certainty, the UK needs to increase its NDCs by 58% [143]. As highlighted by the Climate Action Tracker, if all countries were to follow the UK's approach, global warming could reach almost 3 degrees Celsius. Thus, the UK is failing to do all it can to uphold child rights in Malawi and other lower-income countries, moreover, the UNCRC ruling means that the UK can be held responsible for this. To further improve, the UK needs to develop heating (which currently contributes the most to the UK carbon footprint) that is carbon neutral, improve insulation, reassess how carbon emissions are measured, reassess land use, and employ renewable energy instead of fossil fuels.

The UK already has the capacity to meet energy demands with 100% renewable wind energy now and in 2050, yet 78% of energy is still supplied by fossil fuels [144]. It is vital people have reliable energy access; however, energy needs to be sourced from renewables which are sustainable. To do this, the UK needs to end fossil fuel subsidies and increase subsidies and investment in renewables. This will promote the use of renewables by consumers and can be used to alleviate fuel poverty, this in turn will signal to the market there is a demand. Similarly, the UK needs to put a price on carbon and other greenhouse gases either through a tax or a cap-and-trade system [5]. Research shows paying for carbon moves people away from carbon intensive goods and services to lower carbon alternatives [145].

## MNCs and banks

Only five UK banks have pledged to reach net zero by 2050 and only one UK bank has set actual targets. UK banks are still invested in companies that are expanding fossil fuel extraction, and investment has increased since the Paris Agreement was signed. Divesting has been impactful in the past, for instance, in helping to end apartheid and combatting advertising from the tobacco industry [146]. However, these cases were less complex; the entire world economy is reliant on fossil fuels. The impact of divestment on the fossil fuel industry has been difficult to measure and in the short term there could be a financial impact, but due to demand, less conscientious investors and banks are willing to fill any financial gaps [146]. Even fossil fuel industries that reduce their extraction in response to public pressure may simply have the position filled by smaller companies less open to public scrutiny. Nevertheless, findings suggest that portfolios that do divest and instead invest in clean energy are better performers than those with fossil fuels [147]. The question of whether investors and banks who stay engaged with fossil fuel companies can yield results is debateable. It relies on an influential relationship, as well as the appropriate knowledge and aims of investors and banks [146].

However, other important non-tangible effects from the divestment campaign are possible, for instance, universities that have engaged in the process may have an impact by stigmatising the fossil fuel industries themselves and increasing public awareness. This may, in turn, influence political policies, which are crucial to combat climate change. Merino-saum et al. suggest if the process is reasonable and proportional while remembering the end goal, the divestment movement can positively combat climate change [148].

Whilst tackling the supply side of the equation is vital, it is also important to tackle the demand side, as fossil fuels have driven development, technology, social and economic progress and supplied livelihoods for many. They are energy rich, cheap, and reliable, and their by-products can be used to create useful materials including some medicines. A lack of reliable energy would subject people to a life of poverty and reliance on solid fuel sources, such as

wood, which cause indoor air pollution [149]. This is especially important in the context of lower-income countries, such as Malawi, which need more scope to increase emissions to grow until renewable alternatives are implemented, while globally aggregate emissions need to reduce.

Affordable renewable energy is needed to aid this development and meet global demand, and thorough concerted behavioural change that leads to sustainable use of natural resources is also essential [97]. This must include impact on biodiversity, deforestation and pollution of our oceans [146]. Integrating green economy measures which consider the interface between the economy, environment and social factors is pivotal for sustainability and necessary for evaluating the true cost of services and products [148].

This may in turn help combat some of the harms caused by MNCs. It is important to review the environmental impact of UK MNCs because some of their emissions are overseas which are not accounted for in the UK net zero strategy. This study highlights environmental harms from UK MNCs operating in Malawi. These harms are rights issues; they threaten food security and a clean and safe environment that children have a right to. A threat to these rights, in turn, undermines the right to health [63]. Indeed, there are examples of Malawians challenging UK MNCs through the court of law including British American Tobacco. These cases were taken on by a UK legal firm and are based on MNCs failure to respect the human rights of Malawian citizens [150,151]. Strengthening these remediation channels for Malawians, especially with regards to climate change, is a positive move, such cases are already happening in Europe [30].

Adverse environmental impacts from tobacco include cultivation harms, such as pesticide use, deforestation, biodiversity loss, and green tobacco sickness. Social harms include increased food insecurity and increased poverty. The production process releases toxic compounds as well as greenhouse gases; the consumption process is not only harmful to the consumer but also to non-consumers and the wider environment. The UK has signed and ratified the WHO Framework Convention on Tobacco Control and agreed to protect the environment and the health of persons in relation to tobacco cultivation and manufacture. We believe, given two of the tobacco firms in Malawi are headquartered in the UK, the UK government could do more to regulate these MNCs. For example, prevention of deforestation and ensuring safe labour conditions on farms where the tobacco is cultivated. In addition, MNCs must ensure there is no exploitation which is a requirement of the UK's Modern Slavery Act. Indeed, as mentioned British American Tobacco is being charged in the UK with breaching the Modern Slavery Act in its operations in Malawi; the court case is ongoing [150].

Similarly, the production of sugar in Malawi uses pesticides, is water intensive, and burning, that is part of the cultivation process, causes air pollution. These wider environmental and social harms need to be reflected in the cost of the end product, which a green economy approach would do. The UK is obliged to protect human rights from harms by businesses and therefore needs to mandate comprehensive scope three emission reporting from MNCs and support them to reach net zero. Equally important, the UK must mandate due diligence and impact assessments from MNCs to do no environmental or social harm [63,152]. MNCs should work closely with national and local governments and citizens of Malawi to ensure mitigation of harms are effective and assess where they can have positive impact through their activities. Increased transparency is needed regarding which MNCs are headquartered in the UK.

The ambitions of MNCs can also influence government policies and evidence suggests they engage in extensive lobbying. However, much of this occurs at a supranational level or with selected national governments. This highlights that MNCs can affect government policies either directly or through international organisations [153]. Similarly, banks engage in

extensive lobbying on government policies although they are only successful in 10% of cases [154]. This is important as conflicts of interest can arise between MNCs and banks, on the one hand, and policies that tackle climate change and tax abuse, on the other.

This paper finds that the impacts of climate change on health in Malawi are influenced through multiple different pathways by the UK. These pathways from Fig 1 are interlinked, for example, UK government policies on scope three reporting impact how UK MNCs operate in Malawi.

## Strengths and limitations

This paper offers a unique review on how higher-income countries impact health in lower-income countries from a climate change perspective. It gives an overview of pathways infrequently explored, highlighting key areas that need to be addressed by policy makers in a globalised world.

As mentioned earlier, it was not possible to obtain a comprehensive reliable list of MNCs headquartered in the UK that work in Malawi. This means some key MNCs relevant to this paper will have been excluded. Further, the assessment of projects implemented through UK financing relied solely on secondary evaluations and analysis. Given the complexity of the topic, a systematic review was not possible but future work could explore the different pathways in Fig 1 in more depth.

## Conclusion

Children have a right to health and rights that impact health, including a clean and safe environment, food, sanitation, water, and education. These are recognised in multiple international human rights treaties. Child health in Malawi is already significantly and negatively impacted by climate change and the UK is negatively contributing to this through multiple pathways.

The UK must contribute its fair share of climate finance to lower-income countries and the majority of this should be for adaptation. The UK can act as an advocate for lower-income countries with global institutions and promote a fair and equitable process for allocation of funding ensuring that the most vulnerable countries, historically the least polluting, receive their share of climate finance. Domestically, the UK government should reduce loopholes which facilitate tax abuse and support efforts underway towards an international tax framework within the United Nations.

The UK government must view the climate change crisis as a human rights crisis and remember that it can be held responsible for the negative impact of its carbon emissions on children's rights both within and outside its territory. Equally, it can be held responsible for the cross-border impact of its tax policy on the rights of children. The UK has a duty to review its governmental policies on climate change from a human rights perspective, and it has a duty to collaborate with the international community to protect the environment [155].

To mitigate further harm to lower-income countries like Malawi, the UK needs ambitious climate change policies as recommended by the CCPI. This includes a price on carbon, a review of land use, and improvements to insulation and heating. Additionally, there must be wider employment of renewables and the inclusion of overseas and aviation emissions in its plan to reach net zero. The UK also needs to increase its research and development funding into climate change solutions five-fold.

States have an obligation to protect citizens' rights from the harms done by businesses and banks, and a duty to ensure that all persons, especially those in vulnerable situations, can adapt to climate change. The UK needs to take steps to support its MNCs and banks to reach net

zero and prevent further environmental and social harms. It must view this as its duty to uphold human rights to prevent foreseeable harms caused by climate change both domestically and across the world.

## Author Contributions

**Conceptualization:** Eilish Hannah, Rachel Etter-Phoya, Bernadette O'Hare.

**Writing – original draft:** Eilish Hannah, Bernadette O'Hare.

**Writing – review & editing:** Eilish Hannah, Rachel Etter-Phoya, Marisol Lopez, Stephen Hall, Bernadette O'Hare.

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
