## [Decision Letter · Decision Letter 0]

6 Dec 2022

PGPH-D-22-01397

Impact of higher-income countries on child health in lower-income countries from a climate change perspective. Using the UK and Malawi as a case study.

Dear Dr. Hannah,

Thank you for submitting your manuscript to PLOS Global Public Health. After careful consideration, we feel that it has merit but does not fully meet PLOS Global Public Health’s publication criteria as it currently stands. Therefore, we invite you to submit a revised version of the manuscript that addresses the points raised during the review process.

As indicated by the reviewers, the focus and the organisation of the paper needs to be improved in order to improve readability and have a meaningful, evidence-based, take-home message. I would recommend maintaining the original focus on UK policies on Malawi children health, as stated in the title.

We look forward to receiving your revised manuscript.

Kind regards,

Valentina Gallo

Academic Editor

Journal Requirements:

1. Our staff editors have determined that your manuscript is likely within the scope of our Climate Change and Human Health Call for Papers. This editorial initiative is headed by a team of Guest Editors for PLOS GPH: Renzo Guito (St. Luke's Medical Center College of Medicine, Philippines) and Tolu Oni (University of Cambridge) as well as the Guest Editor for PLOS Climate Anna Stewart Ibarra (Inter-American Institute for Global Change Research). The Collection will feature research that addresses all aspects of the intersection between climate and health, from the changing burden of communicable and non-communicable disease to the impacts of extreme events on health systems, as well as research that assesses potential adaptations to build healthier and more resilient societies. Additional information can be found on our announcement page: https://collections.plos.org/call-for-papers/climate-change-and-human-health/. 

If you would like your manuscript to be considered for this collection, please let us know in your cover letter and we will ensure that your paper is treated as if you were responding to this call.  Please note that being considered for the Collection does not require additional peer review beyond the journal’s standard process and will not delay the publication of your manuscript if it is accepted by PLOS GPH. If you would prefer to remove your manuscript from collection consideration, please specify this in the cover letter.

2. Please send a completed 'Competing Interests' statement, including any COIs declared by your co-authors. If you have no competing interests to declare, please state "The authors have declared that no competing interests exist". Otherwise please declare all competing interests beginning with the statement "I have read the journal's policy and the authors of this manuscript have the following competing interests:"

3. Please amend your detailed Financial Disclosure statement. This is published with the article. It must therefore be completed in full sentences and contain the exact wording you wish to be published.

4. Please provide separate figure files in .tif or .eps format only and remove any figures embedded in your manuscript file. Please also ensure that all files are under our size limit of 10MB.

Additional Editor Comments (if provided):

Reviewers' comments:

Reviewer's Responses to Questions

**Comments to the Author**

1. Does this manuscript meet PLOS Global Public Health’s publication criteria? Is the manuscript technically sound, and do the data support the conclusions? The manuscript must describe methodologically and ethically rigorous research with conclusions that are appropriately drawn based on the data presented.

Reviewer #1: Partly

Reviewer #2: Partly

2. Has the statistical analysis been performed appropriately and rigorously?

Reviewer #1: N/A

Reviewer #2: N/A

3. Have the authors made all data underlying the findings in their manuscript fully available (please refer to the Data Availability Statement at the start of the manuscript PDF file)?

Reviewer #1: Yes

Reviewer #2: Yes

4. Is the manuscript presented in an intelligible fashion and written in standard English?

Reviewer #1: Yes

Reviewer #2: Yes

5. Review Comments to the Author

Reviewer #1: This manuscript addresses an important, topical, and necessary subject that is suitable for PLOS Global Public Health journal. It is evident the bilateral relationships between higher and lower income countries concerning climate change is an understudied area in current literature and this primary research conducted by the authors has potential to be an important addition to the field.

The introduction/background section builds a strong and well evidenced argument of the impact climate change has on children directly and indirectly and how it threatens their human rights referencing appropriate charters and treaties to build this case. However, a stronger case in the background could be built surrounding the inequalities in contribution to climate change and their experiences between high and low- and middle-income countries, a main claim of this paper, by providing more context by drawing on the structural roots of these inequalities experienced such as colonialism, imperialism, global governance, and direct and indirect spheres of influence. This would enable a more critical perspective to why there is a need to hold high income countries accountable and the need for them to support low- and middle-income countries to adapt to and mitigate climate change. This lens and line of thought would also be beneficial to examine and explain UK and Malawi’s relationship further with a critical lens providing further rationale for the reader for this study and why the UK and Malawi were specifically chosen as case studies.

The research question and aim of this study are clearly defined enabling the reader to engage with the scope of this research. However, the methodology section of this paper is its weakest section. The authors share a review of the literature was conducted with search engines and search criteria. However, no further details are provided regarding the search itself such as date of search and how the search was conducted for example the number of authors of involved in the search itself. Further no details are shared regarding the nature of literature review conducted, and how the data was extracted and analysed and synthesised. This information is important to establish the methodological rigour of the study.

The results section is a lengthy and detailed section. However, it would be recommended the results section begins with an overview of the literature collected and consequently used to derive the results. A description of each literature/data source used to include type of literature, source, authors etc is recommended to be included in the manuscript itself or in the appendix to make the readers aware where of all the data underlying these findings comes from. It is evident the results section is structured around Figure 1 shared in the background section, however more detail again could be shared regarding the choice and context behind this framework.

The results section shares in depth Malawi’s climate change crisis and the impact on children’s health across numerous dimensions including air pollution, ecosystem disruption, perhaps the interconnection of these consequences could be examined further for example using a planetary health lens. An important section on adaption and climate finance is included. The table provided could be further expanded to include details of who is funding the fund mentioned, date and timeline of project where applicable and if they were funded through grant, co-finance, or both. This table could perhaps also include data on international finance mentioned in section 2.2 The UK acting via international organisations with regards to climate finance and the two sections could be combined. A good comparison of UK’s contribution to climate finance with other nations and the benefits and shortfalls of present projects, which could also be tabulated. A key and strong section of the results is the impact on UK climate policy on mitigation and children particularly the examination of the role of MNCs and the shortcomings in their report of carbon emissions. This section evidences the breadth of actors involved and their power and influences.

A well written discussion section that summarises key findings, and argues the need for adaption, research and development, mitigation and the role UK needs to play in addressing Malawi’s climate crisis and the need to treat it as human rights crisis. To elevate the discussion of the results further, it is recommended the results addressing each research question are synthesised further through the exploration of how the pathways indicated in the figure interact and influence one another. Additionally, the discussion section would benefit from an evaluation of the strengths and limitations of this research study. The manuscript is clear and coherent however there are omissions in grammar in places, the table has no title/caption and there are inconsistencies in referencing style.

Overall, an important piece of research based on the premise of children’s right to health and human rights which are impacted by climate change using Malawi as a case study. It can be seen UK’s contribution to address this issue needs to advance in many aspects as highlighted in the conclusion. However, this study fails to share sufficient details regarding what data and how this data used to derive these findings and the subsequent conclusions, a key area requiring revision.

Reviewer #2: Interesting idea and paper. It s too long, though, and has a lot of paragraphs repeating the same general arguments.

There is no enough evidence for the relationship especially -as it is noted in the title - of UK policies on Malawi child health.

Needs a major revision with deeper focus on arguments in order to support this specific relationship. Another option would be to change the title and write a general analysis on this interesting topic based on global policies and data.

6. PLOS authors have the option to publish the peer review history of their article (what does this mean?). If published, this will include your full peer review and any attached files.

**Do you want your identity to be public for this peer review?** For information about this choice, including consent withdrawal, please see our Privacy Policy.

Reviewer #1: No

Reviewer #2: No

---

## [Decision Letter · Decision Letter 1]

28 Nov 2023

Impact of higher-income countries on child health in lower-income countries from a climate change perspective. A case study of the UK and Malawi.

PGPH-D-22-01397R1

Dear Ms Etter-Phoya,

We are pleased to inform you that your manuscript 'Impact of higher-income countries on child health in lower-income countries from a climate change perspective. A case study of the UK and Malawi.' has been provisionally accepted for publication in PLOS Global Public Health.

Best regards,

Julia Robinson

Executive Editor

Reviewer Comments (if any, and for reference):

Reviewer's Responses to Questions

**Comments to the Author**

1. If the authors have adequately addressed your comments raised in a previous round of review and you feel that this manuscript is now acceptable for publication, you may indicate that here to bypass the “Comments to the Author” section, enter your conflict of interest statement in the “Confidential to Editor” section, and submit your "Accept" recommendation.

Reviewer #1: All comments have been addressed

2. Does this manuscript meet PLOS Global Public Health’s publication criteria? Is the manuscript technically sound, and do the data support the conclusions? The manuscript must describe methodologically and ethically rigorous research with conclusions that are appropriately drawn based on the data presented.

Reviewer #1: Yes

3. Has the statistical analysis been performed appropriately and rigorously?

Reviewer #1: N/A

4. Have the authors made all data underlying the findings in their manuscript fully available (please refer to the Data Availability Statement at the start of the manuscript PDF file)?

Reviewer #1: Yes

5. Is the manuscript presented in an intelligible fashion and written in standard English?

Reviewer #1: Yes

6. Review Comments to the Author

Reviewer #1: The authoring team have taken on board the comments and feedback made from the reviewers strengthening the article particularly the background section through its wider structural context and the methodology by giving clearer indication of how the literature was selected. The discussion and results have also further synthesised and linked.

7. PLOS authors have the option to publish the peer review history of their article (what does this mean?). If published, this will include your full peer review and any attached files.

**Do you want your identity to be public for this peer review?** For information about this choice, including consent withdrawal, please see our Privacy Policy.

Reviewer #1: No
